# Genetic evidence for the association between COVID-19 epidemic severity and timing of non-pharmaceutical interventions

Manon Ragonnet-Cronin [1✉], Olivia Boyd[1,3], Lily Geidelberg[1,3], David Jorgensen[1,3], Fabricia F. Nascimento[1,3], Igor Siveroni [1,3], Robert A. Johnson[1], Marc Baguelin[1], Zulma M. Cucunubá [1], Elita Jauneikaite [1], Swapnil Mishra [1], Oliver J. Watson [1], Neil Ferguson [1], Anne Cori[1], Christl A. Donnelly [1,2] & Erik Volz [1]

Unprecedented public health interventions including travel restrictions and national lockdowns have been implemented to stem the COVID-19 epidemic, but the effectiveness of non-pharmaceutical interventions is still debated. We carried out a phylogenetic analysis of more than 29,000 publicly available whole genome SARS-CoV-2 sequences from 57 locations to estimate the time that the epidemic originated in different places. These estimates were examined in relation to the dates of the most stringent interventions in each location as well as to the number of cumulative COVID-19 deaths and phylodynamic estimates of epidemic size. Here we report that the time elapsed between epidemic origin and maximum intervention is associated with different measures of epidemic severity and explains 11% of the variance in reported deaths one month after the most stringent intervention. Locations where strong non-pharmaceutical interventions were implemented earlier experienced much less severe COVID-19 morbidity and mortality during the period of study.

[1] MRC Centre for Global Infectious Disease Analysis and the Department of Infectious Disease Epidemiology, Imperial College London, London, UK. [2] Department of Statistics, University of Oxford, Oxford, UK. [3] These authors contributed equally: Olivia Boyd, Lily Geidelberg, David Jorgensen, Fabricia F. Nascimento, Igor Siveroni. ✉email: manonragonnet@imperial.ac.uk

To prevent the spread of the SARS-CoV-2 virus, countries have implemented unprecedented measures ranging from school closures and travel bans to large-scale lockdowns[1]. Because of the immense economic and social consequences of such interventions, the utility of travel restrictions and lockdowns have been questioned by the public and media. Mathematical models can help determine whether these non-pharmaceutical interventions (NPI) have been effective in reducing transmission. Previous investigations based on mathematical models have shown that travel restrictions in Wuhan delayed arrival of the virus into other Chinese cities by around 3 days, and early implementation of control measures was associated with lower incidence[2]. Similarly in Germany, changes in epidemic growth rates are correlated with the timing of interventions[3]. In order to assess the impacts of NPIs across European countries, Flaxman et al. estimated the rate of epidemic growth over time from reported COVID-19 deaths and pooled information across European countries[4]. The impact of NPIs was significant across all countries, reducing the reproduction number below 1, and reducing the number of deaths compared to a scenario with no intervention. Differences in case reporting complicate international comparisons, but as the scale of epidemics during the first wave varied so drastically across countries, such comparisons are essential. Dye et al.[5] found that the 100-fold difference in cumulative COVID-19 deaths across European countries was not attributable to lower rates of transmission and epidemic growth, but rather due to differences in dates of national lockdown.

International comparisons are more complex due to greater variability in epidemiological surveillance and epidemic growth rates. Furthermore, while sustained transmission took place across most of Europe from around mid-February[6], global comparisons are complicated by the fact that the virus was introduced in different places at very different times. Epidemiological models are typically fitted using data on reported infections, deaths, and seroprevalence surveys. These provide estimates of the basic reproduction number, $R_0$, the average number of onward transmissions per case in a susceptible population; and the time-varying effective reproduction number, $R_t$. However, because a high proportion of SARS-CoV-2 infections are not detected and seroprevalence surveys have only recently been carried out, parameterising models for epidemics in February through April when large-scale NPIs were enforced is not straightforward. Critically, it is usually not understood how long SARS-CoV-2 circulated in populations prior to detection, which is essential information for comparing timing of NPIs between locations.

Viral genetic data provide an alternative source of information for understanding when epidemics originated and how quickly they grew during the first wave. Analysis of SARS-CoV-2 genetic sequences can help distinguish between imported and local transmission; and inclusion of sample dates allows for further time-resolution of epidemic dynamics, yielding dates of viral introduction into specific regions. Phylodynamic analysis uses genetic data to parametrise epidemiological models, for example estimating $R_t$ directly from viral sequences[7]. Analyses of viral sequences have demonstrated that transmission of SARS-CoV-2 went undetected within the USA from mid-January[8–11] and many epidemics were seeded between states rather than as a result of international travel. In contrast in Guangdong, China, the majority of new diagnoses appeared to be imports, demonstrating the effectiveness of surveillance and interventions in interrupting community transmission[12].

In the present analysis, we reconstruct the epidemic trajectories of SARS-CoV-2 outbreaks for locations across the world for which genetic data were available, to evaluate the scale of epidemics. We estimate the time of viral introduction into each region and calculate time to lockdown for each region. We then determine whether reported and estimated epidemic sizes relate to the duration of times to lockdown.

## Results

**Time of viral introductions**. We identified 57 locations (24 in Europe, 20 in North America, 5 in the Middle East, 6 in Asia, 1 in South America and 1 in Africa) meeting inclusion criteria where dates of public health interventions could be obtained and where publicly available SARS-CoV-2 sequences enabled phylodynamic analysis using non-parametric methods (Fig. 1). For each site, we extracted dates of the first and the most stringent NPI (maximum NPI, usually lockdown) from the Oxford COVID-19 Government Response Tracker[1]; as well as case and death counts and census population sizes for each location (Supplementary Table 1 and Supplementary Fig 1). In total, 29,163 SARS-CoV-2 sequences were analysed, in regional datasets ranging from 23 sequences (Innsbruck, Austria) to 677 sequences (Denmark; Supplementary Fig 2). Sample dates ranged from 2020-01-08 to 2020-05-30.

For each region, we estimated the timing of viral introductions through time-resolved phylogenetic analysis and parsimony reconstruction. Epidemic seeding was a continuous process with all regions showing evidence for more than one epidemic origin (Fig. 2). The earliest seeding (lineage importation) events among the 57 sites took place in January and in most locations seeding continued with a maximum frequency occurring in March before maximum NPI and travel restrictions were implemented in most locations. As a phylogenetic estimate of the time of viral introduction, we use the central epidemic seeding time (CEST), the mean time of viral introductions weighted by the number of samples descended from each viral introduction. Sensitivity analyses utilising other definitions of viral introduction are presented in Supplementary Information. Time delay from viral introduction (CEST) to maximum NPI for each location varied from −4 days (meaning the NPI took place 4 days before CEST) to 2 months (Fig. 3 and Supplementary Fig 3).

**Association between time to non-pharmaceutical interventions and epidemic severity**. We constructed a series of regression models with time from CEST to maximum NPI (in days) as predictor variable and a measure of the scale of the epidemic (deaths one month after the NPI) as the outcome. As estimated and extracted variables contained some level of uncertainty, we used Deming regression models, which account for measurement error in both predictor and outcome variables and allow for weighting of observations based on the precision of their estimates[13–15]; as well as linear regression models, which allow for inclusion of multiple predictors. Among the 57 sites, CEST to maximum NPI was significantly associated with the number of deaths reported at each site 1 month following the time of maximum NPI (Fig. 4, Supplementary Data 1 and Supplementary Table 1). Time from CEST to maximum NPI was predictive of the number of deaths 1 month later in the Deming regression ($p = 0.0031$; Supplementary Fig 5) and in the univariate linear model ($R^2 = 0.11$, $p = 0.011$). An additional 14 days of transmission before maximum NPI was associated with a 2.91-fold (95% CI: 1.35–6.27) increase in deaths 1 month after maximum NPI in the Deming model and a 2.00-fold (95%CI: 1.19-3.32) increase in the univariate regression. Meanwhile, time from the tenth reported case was not significant in either model ($p = 0.55$ and $p = 0.8$, respectively). Census population size of each location was not significant in a univariate model predicting deaths ($p = 0.08$).

**Association between time to non-pharmaceutical interventions and viral effective population size**. For each region ($n = 57$), we

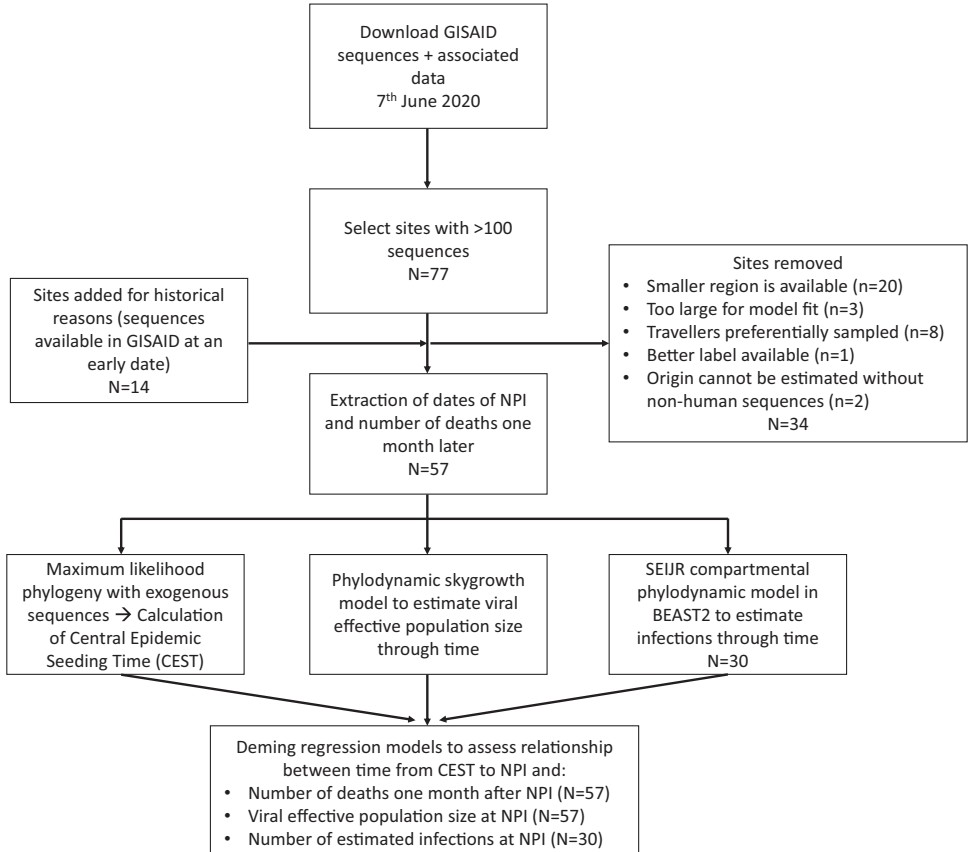

**Fig. 1 Flow chart showing the analysis pipeline.** The number of geographical sites included at each stage of analysis is noted. Reasons for exclusion for the 34 sites removed are detailed in the Supplementary Information (section 1B). SEIJR Susceptible-Exposed-Infected (IJ)-Recovered, CEST central epidemic seeding time, NPI: non-pharmaceutical intervention. GISAID database https://www.gisaid.org/.

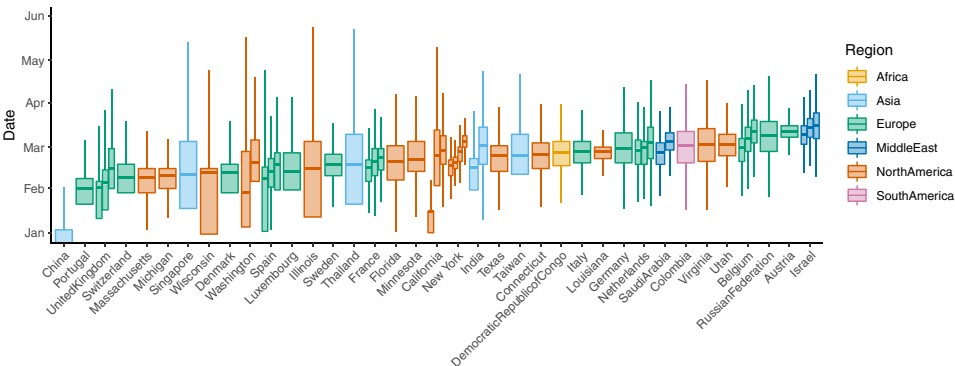

**Fig. 2 Distribution of phylogenetically inferred times of importation events for the 57 sites included in our analysis, also referred to as seeding time.** Ancestral states are reconstructed onto the phylogeny as exogenous or within the region of interest. The midpoint between an ancestral node which is exogenous to the region of interest and a node which is within the region of interest is the time of importation. Local epidemics can be seeded by many importation events. Central box-plot lines represent the central epidemic seeding time (CEST). Note that the CEST does not represent the earliest viral introduction but rather the mean time of viral introduction weighted by the number of samples descended from each viral introduction. Boxes represent the interquartile range of the distribution and whiskers indicate the 25th and 75th percentiles of the distributions, derived from $n = 2500$ data points (100 time-resolved phylogenies for each location, with ancestral states reconstructed 25 times on each). Ranges reflect the distribution of importation dates, rather than uncertainty around an estimate of seeding times. Sites from the same country or US state are grouped together and sites are ordered based on the first CEST for that country or US state. Width serves only to compress multiple sites from the same country or US state.

applied a skygrowth model[16,17] (version 0.3.1) to estimate viral effective population size through time and growth rates of effective population size. Under appropriate conditions, effective population size can be used as a proxy statistic for epidemic prevalence[7]. Viral effective population size at maximum NPI was similarly associated with time to maximum NPI in the Deming

($p < 0.0001$; Fig. 3, Supplementary Fig 5) and the univariate regression ($R^2 = 0.32$, $p < 0.0001$; Fig. 4, Supplementary Table 1). An additional 14 days of transmission before maximum NPI was associated with a 2.18-fold (95% CI: 1.52–3.13) increase in effective population size at time of maximum NPI in the Deming model and 2.08-fold (95% CI: 1.54–2.83) increase in a univariate

linear model. In a univariate model, census population size was predictive of viral effective size ($R^2 = 0.15$, $p = 0.0046$). In a multivariable model ($R^2 = 0.38$), time from CEST to maximum NPI remained a significant predictor ($p = 0.0005$) but census population size did not ($p = 0.09$). The number of deaths and viral effective population size were highly correlated (Pearson's $r = 0.58$, $p < 0.0001$; Supplementary Fig 4). Note that the models for estimating CEST and models for estimating viral effective population size were fit to data independently (although run on the same set of data) and therefore association between these results is not due to a circularity in inference.

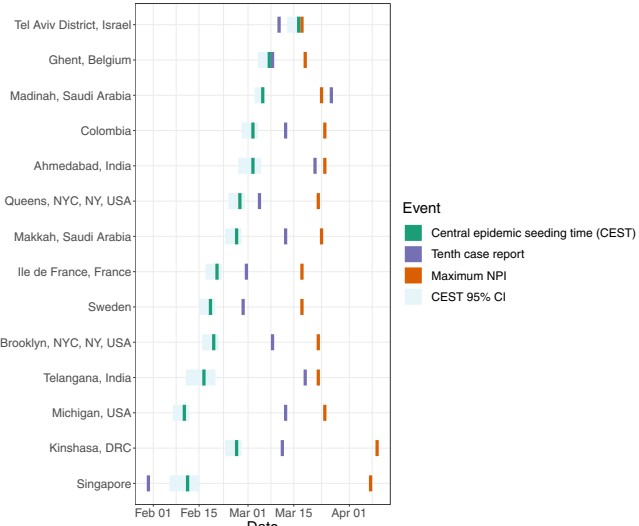

**Fig. 3 Dates of central epidemic seeding time (CEST), tenth case report and maximum non-pharmaceutical interventions (NPI) for 14 of 57 sites included in our analysis.** Sites are ordered by the duration of time between CEST and maximum NPI. Note that the CEST does not represent the earliest viral introduction but rather the mean time of viral introduction weighted by the number of samples descended from each viral introduction; thus it can be preceded by the tenth diagnosis if early diagnoses are not phylogenetically related to later infections (e.g., Singapore). CEST is shown with 95% confidence intervals (CI). 95% CI are rounded to the nearest day and therefore do not always look symmetrically distributed around CEST. We selected up to three sites from each world region for which sites were available (Europe, North America, Middle East, Asia, Africa) based on their having the highest death counts in the region. Supplementary Fig 3 shows these dates for all 57 sites.

**Epidemic parameters inferred through phylodynamic analysis.** Across all sites and using non-parametric methods, we estimated the mean epidemic doubling time to be 3.68 days (IQR: 2.45–5.76). A subset of sites were analysed in BEAST2 v6.1 using model-based phylodynamic methods[18,19] to estimate the effective reproductive number and the number of infections through time (Supplementary Fig 6). Similar results were obtained among those 30 sites: epidemic doubling time was 3.47 days (IQR: 3.09–4.83, see Supplementary Information). Estimates of $R_0$ varied between 1.53 (in Makkah, Saudi Arabia) and 4.18 (in New Orleans, Louisiana, USA; Supplementary Table 2; Supplementary Fig 4).

To corroborate inference of effective reproduction numbers and epidemic size from genetic data, we examined the relationship between inferred quantities and reported numbers of cases as well as independent information about human mobility patterns (see Supplementary Information). We found that changes in the inferred reproduction number through time, $R_t$, corresponded with changes in Google human mobility metrics for 29 of 30 locations where these data were available (Supplementary Table 2; Supplementary Fig 7). Large reductions in human mobility metrics consistently correspond with periods when $R_t$ decreases.

## Discussion

Among 57 geographical sites sampled across five continents, we found that time from SARS-CoV-2 introduction to time of lockdown (or maximum NPI in locations that never underwent a full lockdown) was significantly associated with the severity of the epidemic in each location during the first wave. This result is consistent with analyses[20] conducted thus far separately in China[2,21] and Europe[3–5] and demonstrates the importance of cryptic transmission before testing and interventions were implemented in exacerbating epidemic severity. Notably, the time between detection of the tenth case at each site and the maximum NPI was not predictive of the number of deaths. This latter finding indicates that many locations experienced long and variable periods of cryptic transmission before epidemics were detected[8–11]. Based on our numerical analysis, and acknowledging highly variable outcomes across sites, implementing a strong NPI such as national lockdown 2 weeks earlier would have approximately halved cumulative deaths in the period immediately following lockdown, on average. The time from viral introduction to the first NPI was also associated with severity (see Supplementary Information).

A previous comparison across European countries found that earlier lockdown dates were associated with fewer deaths, and that countries with fewer COVID-19 deaths had fewer inhabitants[5].

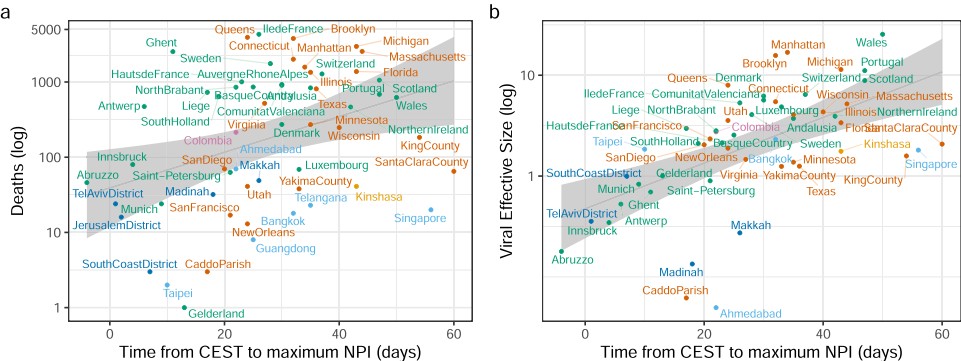

**Fig. 4 Relationship between epidemic severity and viral effective population size, and the estimated time to maximum NPI. a** Cumulative deaths within a month following max NPI versus time to max NPI. **a** Viral effective population size (estimated at max NPI) versus time to max NPI. Colours are different for each world region (see Supplementary Fig 4). The grey line indicates the linear regression line (mean fitted values) and 95% confidence interval. CEST central epidemic seeding time, NPI non-pharmaceutical intervention.

An association between population size and the number of deaths could result from depletion of susceptibles, limiting onward spread in places with smaller populations. However, serological surveys and patterns of per-capita death do not support the hypothesis that herd immunity has been reached anywhere in Europe[22]. In our global analysis, census population size was predictive of the number of estimated infections, viral effective population size and the number of deaths in univariate models. But when controlling for the time from CEST to lockdown, population size no longer had an effect, which is consistent with the hypothesis that herd immunity has not substantially limited transmission. The timing of viral introductions and all measures of epidemic scale were inferred independently.

International comparisons of NPI effectiveness have been complicated by widely varying testing strategies in different locales and most epidemiological models are highly reliant on reported COVID-19 diagnoses and deaths. Our ability to run analyses on such a wide range of locations derives from the fact that our model is parameterised entirely by genetic data. We have shown that this approach produces accurate estimates of the numbers of infections in simulations (Supplementary Information), and among the study sites, we found that the number of estimated infections was highly correlated with the numbers of COVID-19 diagnoses and deaths. Nonetheless, we acknowledge the limitations of reported death data: reliability will vary based on location, and we only included a single time point because of difficulties in extracting such data. Excess mortality figures may provide less biased estimates, but have not been calculated at the level of resolution required for our study[23]. Viral effective population size is not always a quantity directly relatable to the number of infections[7]; however, our analyses support the use of viral effective population size at maximum NPI as a proxy for SARS-CoV-2 epidemic size at that time. Where possible, we compared our estimates to those modelled or measured elsewhere. Estimates of $R_0$ and $R_t$ were in line with those previously reported for the same locations[24–28] and $R_t$ decreased synchronously with reductions in human mobility, as previously demonstrated for $R_t$ estimates derived from traditional epidemiological models[29–31]. Dates of introduction into Europe aligned with previous reports demonstrating sustained transmission from mid-February[6].

A limitation of our analysis is that the Bayesian MCMC for our SEIJR phylodynamic model did not converge for all the locations. Lack of convergence can occur because of problematic datasets, in which samples may for example not be collected at random; because the model is mis-specified; or because one of the model assumptions is violated, for example, the population is not randomly mixing. Where possible, we addressed these concerns by excluding sites known to have prioritised sequencing from travellers[32,33] or contact tracing and by focusing on smaller geographical units, such as cities and small regions where within-sample geographic structure is less biasing. Further optimisation of some parameters, such as the parameter for transmission overdispersion could have improved estimates, as has been recently demonstrated by Miller et al.[33], but we chose to keep this parameter constant across sites to facilitate meta-analysis. Changes in transmission rates close in time to the last sample are difficult to detect using genetic data[16] and in our SEIJR model, the number of estimated daily infections tended not to decrease or stabilise, despite external evidence that reported cases were levelling off.

The non-parametric phylodynamic analysis allowed us to include sites for which data were available but the Bayesian MCMC did not converge[16,17], as well as to examine sensitivity of results to choice of modelling framework. Results echoed those from the compartmental phylodynamic model. Viral effective population size at maximum NPI was associated with deaths 1 month after maximum NPI and with time to maximum NPI.

One limitation of the skygrowth phylodynamic model is that it does not explicitly consider imports into the region of interest. As sequences from the region are generally derived from multiple distinct importation events the method may incorrectly estimate viral effective population size for a region, especially early on. However, as an epidemic expands in a region and more infections are attributable to endogenous transmission, the approximations in this method improve. Lineage imports were explicitly parameterised in the SEIJR model and results were consistent across both methodologies.

In conclusion, we have shown that across five continents, longer delays from viral introduction to lockdowns were associated with more infections at lockdown and more deaths 1 month after lockdown. The association may be causal but an observational study such as ours cannot draw that conclusion. Our study focused on the first wave of the pandemic, however, lifting of interventions and waning compliance have led to subsequent waves of infection across many of the sites included. Our models were calibrated entirely using genetic data and thus provide an independent confirmation of mathematical models calibrated to traditional data sources. These findings emphasise the importance of NPI for decreasing epidemic severity, reinforce previous findings that seroprevalence is below that needed for herd immunity[22] and highlight the risk for re-emergence and continued transmission.

## Methods

**Genetic data preparation and selection of study sites**. SARS-CoV-2 sequences were downloaded from GISAID (http://gisaid.org)[34] on June 7, 2020. Genetic data were cleaned and prepared for time-resolved phylogenetic analysis in R v3.6.1 (Supplementary Information). The GISAID database classifies the geographical origin of each sequence with four levels of resolution: world region, country, division and location, We chose to focus our analysis on sites, regardless of scale, with at least 100 sequences available (although some sites with fewer sequences were analysed, see Fig. 1 and Supplementary Information for details). Thus a site, as included in our analysis, could represent anything from a neighbourhood (e.g., Manhattan) to a country (e.g., Denmark), depending on the level of resolution of geographical data available with sequences from that region. All sites analysed are displayed in Supplementary Fig 2 and listed in Supplementary Data 1. GISAID IDs for all sequences included in the final analyses are listed in Supplementary Data 2. When data were available for sites located within each other (e.g., New Orleans in Louisiana), the smaller geographic unit was preferentially selected to better match the premise of the phylodynamic model (see Supplementary Information and Discussion). Sites known to comprise many sequences from travellers or generated as a result of contact tracing were excluded. Because our model assumes that samples are taken at random from the population, we excluded duplicate sequences from our analyses as a proxy for membership with the same transmission chain. We investigated the effect of this choice through simulation (see Supplementary Information).

**Estimating the time of regional viral introductions**. The timing of viral introductions was estimated through time-resolved phylogenetic analysis and parsimony reconstruction. We included all sequences available for that region as well as all close exogenous sequences (≤2 substitutions away) in a maximum likelihood (ML) tree, built using IQtree[35]. Within the ML tree, we resolved polytomies at random, and estimated rooted time-scaled phylogenies using treedater[36], repeating the procedure 100 times. The mean clock rate of evolution was constrained between 0.00075 and 0.0015 substitutions per site per year[37]. Branch lengths were smoothed by enforcing a minimum number of substitutions per site on each branch and by sampling from the distribution estimated by treedater. Finally, we reconstructed the ancestral state of nodes and dated and counted importation events, repeating the procedure 25 times. The time of the importation event is counted as the midpoint along the branch between the exogenous ancestral node and the most recent common ancestor node within the region. We calculated the weighted mean from the distribution of viral introduction times and call it the CEST. All functions are available and documented within the sarscov2 R package (https://github.com/emvolz-phylodynamics/sarscov2Rutils). Compared to more sophisticated phylogeographic models implemented in BEAST[38], our method has the advantage of greater computational scalability, while still integrating over phylogenetic uncertainty.

**Non-parametric phylodynamic inference**. For each region, we applied a skygrowth model[16,17] (version 0.3.1) to estimate viral effective population size through time and growth rates of effective population size which under appropriate

conditions can be used as a proxy statistic for epidemic prevalence[7]. Time-resolved phylogenies were constructed, as detailed above, with all exogenous sequences removed prior to analysis. Growth rates and viral effective population size were estimated using skygrowth 0.3.1[16] using Markov chain Monte Carlo (MCMC) and 500 thousand iterations for each time tree and using an Exponential ($10^{-4}$) prior for the smoothing parameter. The final results were produced by averaging across 100 time trees estimated for each region. Code to reproduce this analysis is contained in the sarscov2 R package (skygrowth1 function, https://github.com/emvolz-phylodynamics/sarscov2Rutils).

**Transmission model and comparative phylodynamic analysis**. Finally, we utilised a compartmental structured coalescent model in the BEAST2 v6.1 PhyDyn package[18,19] to estimate the effective reproduction number and the number of infections through time from SARS-CoV-2 genetic sequences. This model allowed us to assess the reporting rate for each site, the proportion of estimated infections that were diagnosed on each day. The phylodynamic model is designed to estimate epidemiological parameters from sequence data. The model of epidemic dynamics is based on a susceptible-exposed-infectious-recovered model and is described in detail in Supplementary Information. The compartmental model has been previously described and applied to SARS-CoV-2 sequence data[33,39,40]. Importantly, the model accounts for bidirectional migration between a region of interest and an international reservoir, and splits the infected compartment into categories representing individuals with high or low rates of onward transmission. Earlier modelling efforts in SARS-CoV-1 and SARS-CoV-2 demonstrated that the inclusion of a high transmission rate compartment is crucial to realistically capturing case numbers[40,41]. The ability to accurately reconstruct epidemic dynamics using the Bayesian MCMC inferential framework and phylodynamic model was assessed in a simulation experiment (Supplementary Information). BEAST simultaneously reconstructs a phylogeny and estimates epidemiological parameters. For each site under investigation, we selected up to 150 unique regional sequences from the GISAID alignment, as well as exogenous sequences representing the international reservoir. Fifty exogenous sequences encompassing the full time-range of GISAID samples were selected each time at random as background, and to these we added sequences from GISAID that were ≤2 substitutions away from the sequences in the regional dataset calculated. Pairwise genetic distances were calculated using TN93 (https://github.com/veg/tn93). It is possible that this number of exogenous sequences is insufficient to correctly estimate import rates into our regions of interest; however, this objective was not a focus of the present analysis. For each regional dataset, we then constructed a phylogeny in IQtree. Polytomies were resolved at random ten times, each time generating a new starting tree for the analysis in BEAST2, totalling ten independent chains.

Each of the 10 runs was set up for 20 million steps. Subsequently, log files were examined for convergence in Tracer v1.7.1, problematic runs excluded, and log files and trajectory files were combined and cleaned using the sarscov2 R package (available at http://github.com/emvolz-phylodynamics/sarscov2Rutils).

**Comparison with other sources of data and statistical analysis**. For every site, we extracted dates of lockdown from the Oxford COVID-19 Government Response Tracker[1] (downloaded 20/06/2020). We used the date of measure C6 (shelter-in-place), as classified by the OxCGRT dataset. If the region never underwent a full lockdown, we used the date of school closures or recommendation to work from home, whichever came first (details in Supplementary Table 1). We henceforth name this intervention maximum NPI. We also extracted the dates of the first public information campaign (classification H1) for each site, except for sites in the USA, where no such data were available. For the USA sites, we used the date of the Chinese travel ban on Feb 2, 2020, as this attracted extensive media coverage and would have raised awareness among the general population.

We obtained case and death counts, and census populations sizes, for inclusion in our models and to assess case reporting (sources and raw numbers listed Supplementary Data 1; Supplementary Fig 1). We selected to count cases on the date of maximum NPI as a measure of the scale of the epidemic before that intervention. We used the death count 1 month later to account for the incubation period (95% of patients show symptoms within 12 days[42]) and the period of time from symptom onset to death (95% occur within 19 days[43]).

We then calculated the time from epidemic origin (CEST) to each NPI and the time from the tenth reported case to each NPI for each site and constructed a series of regression models, looking for relationships between this delay and the severity of the epidemic at each site. Deming regression models were utilised to take into account for uncertainty in both predictor and outcome variables and allow for weighting of observations based on the precision of their estimates[13–15]. Variance for predictor and outcome variables was calculated as explained in Supplementary Information. However, Deming regression models do not allow for the inclusion of multiple predictors, therefore univariate and multivariable linear regression were used to assess the relative contribution of predictor variables.

For all sites, we built Deming regression models with time from CEST to maximum NPI (in days) as predictor variable and number of deaths 1 month after the maximum NPI as the outcome. To evaluate if the time that cases were first reported could serve as a proxy for time of epidemic origin, we reran these models changing our predictor variable to time between the tenth reported case and maximum NPI. For the 30 locations where the number of infections was estimated,

we built a model with time from CEST to maximum NPI as a predictor and the estimated number of infections as an outcome. We repeated all three CEST-based models using univariate linear regression, then expanded them into multivariable models, including population size and $R_0$ as predictors. Population size, viral effective population size, estimated infections, reported infections and death counts were log-transformed prior to analysis, to reflect their exponential rate of increase. As a sensitivity analysis, as well as the CEST, we also calculated the 5th and 25th percentiles of the distribution of viral introduction times and recalculated the delay to maximum NPI for each definition, using these variables as predictors in regression models. All analyses were conducted in R v3.6.1.

For sites analysed using the BEAST phylodynamic model, we examined the relationship between mobility data provided by Google (google.com/covid19/mobility) and $R_t$[29,31]. Google mobility data measures daily mobility at the sub-region level, in relative deviations from maximum mobility prior to the WHO pandemic declaration. Mobility data were available from the 13th of January 2020 and up until the date of cut-off for genetic data (June 7th). For each site analysed, we plotted both variables for the time period between 13th of January and the date of the last sample available for that site. We limited our analysis to mobility associated with transit stations only (one of six streams). Daily $R_t$ estimates through time were extracted for the same time period.

**Data ethics**. Google mobility data (google.com/covid19/mobility) are released in strict adherence to Google privacy protocols, from users who have chosen to turn on "Location History". Data are fully anonymized, aggregated at the sub-region level and provided as relative deviations from mobility prior to the WHO pandemic declaration.

**Reporting summary**. Further information on research design is available in the Nature Research Reporting Summary linked to this article.

## Data availability

All sequence data are freely available upon registration with the GISAID database (https://www.gisaid.org). The GISAID IDs for sequences utilised in our final skygrowth and BEAST analyses (including background sequences) are listed in Supplementary Data 2. Dates of lockdown were extracted from the Oxford COVID-19 Government Response Tracker[1] (downloaded 20/06/2020). Google mobility data can be accessed at google.com/covid19/mobility. The mobility data utilised in this manuscript is also available in the Github repository https://github.com/manonr/ncomms/tree/main/googlemob_data. Data sources for case and death numbers are available in the Supplementary Data 1. Source data for reproducibility of our statistical analyses (Deming and linear regression) and plots are also available in Supplementary Data 1.

## Code availability

Code for the simulations is available at https://github.com/thednainus/sarscov2simulations (https://doi.org/10.5281/zenodo.4559446). The example output (a fasta file with sequences dated and classified as internal or exogenous to the region of interest) can be used to make dated trees using treedater (previously released, available as an R package https://cran.r-project.org/web/packages/treedater/index.html). Dated trees are the input for the skygrowth model and the method to estimate seeding times. Code for these methods is available as part of the sarscov2 package v0.1.4 at https://github.com/emvolz-phylodynamics/sarscov2Rutils. Fasta files, as generated by, and available in, the simulation package, are the input for the BEAST2 PhyDyn model v1.3.7. Code for the BEAST2 PhyDyn model, as well as instructions and additional test datasets are available at https://github.com/mrc-ide/PhyDyn.

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

## Acknowledgements

We wish to thank the entire Imperial College COVID-19 Response Team. We acknowledge joint Centre funding from the UK Medical Research Council and Department for International Development (MR/R015600/1). This work is also supported by the National Institute for Health Research Health Protection Research Unit in Modelling Methodology, the Abdul Latif Jameel Foundation and the EDCTP2 programme supported by the European Union. RAJ and EV acknowledge funding from the European Commission (CoroNAb 101003653). EV additionally acknowledges funding from the Wellcome Trust (220885/Z/20/Z). OJW reports grants from UK Foreign Commonwealth and Development Office during the conduct of the study.

## Author contributions

MRC and EV conceived of the analysis, performed analyses and wrote the manuscript. OB, LG, DJ collected data and performed analyses. FFN collected data and wrote code for conducting simulations. IS built and tested software. RAJ, AC, CAD provided support and code for statistical analyses. MB, ZMC, EJ, SM, OJW, NF provided support for data collection and analysis. All authors approved the final version of the manuscript.

## Competing interests

EV has an honorary contract with Public Health England (Sep 2020-present) to conduct work in the Genomic Epidemiology Cell. CAD is a member of Council of the Royal Society and a member of Council of the Royal Statistical Society. As a result, CAD is a trustee of both charities. NF has been an unpaid member in a government organisation (SAGE). OJW reports grants from the UK Foreign, Commonwealth and Development Office, during the conduct of the study. All other authors report no competing interests.
