## [Peer Review File · Nature Communications]

REVIEWER COMMENTS

Reviewer #1 (Remarks to the Author):

Preliminary statement: I first need to acknowledge that I consider that I do not have the specific expertise to judge the quality of the multivariate analyses performed in the present study. [For instance, I don't know if an alternative machine learning approach (such as random forest?) could have been more performant and/or relevant than (linear) regression models to analyse the association between epidemic severity and delay between CEST and maximum NPI]

Ragonnet-Cronin et al. here present an interesting study where they used a phylodynamic pipeline to analyse the impact of the timing of non-pharmaceutical interventions in different regions of the world. The study focuses on crucial and timely epidemiological questions. I however have the following concerns regarding the methodology (part of them are actual questions or aspects of the study that were not entirely clear for me):

- I have to admit that I found difficult to follow the pipeline of the phylogenetic and phylodynamic analyses. The authors could consider adding a preliminary Methods paragraph summarising and numerating the main steps that were performed (or, alternatively, generating a figure summarising that workflow). Just some suggestions...

- non-parametric phylodynamic inference: it seems that the skygrowth approach was applied on time-scaled ML trees that were each time inferred from a set of sequences including sequences sampled from the considered site as well as from closed exogenous sequences. Consequently, and except if I missed something in the procedure, such ML tree does not correspond to a single importation event in the considered site. As the authors did, they can be used to identify importation events into the considered site, but a priori not to estimate the trends in the viral effective population size within the site. In other words, I question the application of the skygrowth approach to estimate site-specific viral effective population size based on a ML trees, ML trees that includes several distinct clades gathering sequences sampled in the considered site, each clade corresponding to a distinct importation event. But again, my apologies if I missed a technical aspect in that step of the pipeline detailed in Supplementary Materials

- if I followed correctly, the CEST will also be impacted by importation events that could have occurred after the start of NPIs, or even close to it (?). If this is case, I am actually wondering if this is actually the good estimate to be compared with the maximum NPI. It could indeed impact the delay between CEST and maximum NPI that the authors aim at comparing with the epidemic severity. In other words, why not only try to focus on early importation events (e.g., important events before, or at least some time before, the maximum NPI)?

Additional comments and questions:

- model-based phylodynamic inference of epidemic size and reproduction numbers: the SEIJR is very interesting. On question though: why only 50 exogenous sequences? This number seems rather low to estimate the number of independent importation events. I know that it was not the primary purpose of that step of the pipeline but it can probably tend to underestimate the number of distinct importation events that can in turn impact the inference of the epidemic size

- in my opinion, a sampling map of selected sites (+ the amount of genomic sequences included by site) is missing and could be very informative

- line numbers (as well as figure number and legend below each Supplementary Figure) would have been useful for the reviewing process

- my general feeling is that some parts of the Supplementary Materials should move to the main

Methods text to improve clarity

- Abstract, "The time elapsed between epidemic origin and maximum intervention is strongly associated with different measures of epidemic severity and explains 11% of the variance in reported month after the most stringent intervention": it is only a matter of vocabulary, but I don't know if "strongly associated" is a good match with 11% (?)
- Introduction, "Comparisons beyond Europe are more complex due to greater variability in epidemiological surveillance and epidemic growth rates": the authors should maybe consider a less Europe-centered version of that sentence ("Comparisons between location from different continents are..." or something like that)
- Discussion, "Notably, the time between detection of the tenth case at each site at the maximum NPI was not predictive of the number of deaths": "..._and_ the maximum NPI...?"
- Discussion - as acknowledged by the authors, "international comparisons of NPI effectiveness have been complicated by widely varying testing strategies in different locales and most epidemiological models are highly reliant on reported COVID-19 diagnoses and deaths. Our ability to run analyses on such a wide range of locations derives from the fact that our model is parameterised entirely by genetic data": I agree with this statement but, if I'm correct, the authors do however use death counts for measuring epidemic severity. In that context, the authors should probably discuss the impact on their analyses of some countries underreporting COVID-19 mortality, something that was for instance assessed by comparing this mortality to over-mortality estimated by comparing 2020 to previous years (as some journalists of the NY Times did)
- Methods, "The timing of viral introductions was estimated through time-resolved phylogenetic analysis and parsimony reconstruction": why using a simplistic parsimony reconstruction method instead of one of the discrete diffusion or structured coalescent models implemented in BEAST2?
- Methods, "For sites analysed using the BEAST phylodynamic model, we examined the relationship between mobility data provided by Google (google.com/covid19/mobility, analysis limited to transit stations only) and Rt by calculating Pearson's correlation coefficient": this section needs further detail to be imported from Supplementary Materials
- Methods, "We chose to focus our analysis on sites with at least 100 sequences available": "sites" is actually a little bit too vague and should be defined once for all somewhere in the text
- Methods, "This model allowed us to assess the reporting rate for each site": what do you exactly mean by "reporting rate"?
- Supplementary Materials, "We dropped 80% of sequences collected from the UK after March 15th to reduce bias due to sampling": how did you select the UK sequences to keep in your analyses? This subsampling procedure should be described in detail and motivated
- Supplementary Materials, "We split our data into subsets (~2,000 sequences)...": did you randomly split your data into subsets?
- Supplementary Materials, "... and to each subset added a set of 500 sequences spanning the time period from the first SARS-CoV-2 sample (24/12/2019) to the last sample date": I'm not sure to understand that step (that is the motivation behind it but also how the 500 sequences were selected and the distributed between the 12/24/19 and the last sample date. Could the authors develop further that part?
- Supplementary Materials, "Viral effective population size and the number of infections were

estimated from a distribution of trees (generated through maximum likelihood bootstrap and Bayesian inference, respectively)": do you mean that you repeated the analyses while using trees inferred as described in "Estimating the time of regional viral introductions" and then also as described in "Non-parametric phylodynamic inference"? (The connection between the different steps of the pipeline is not always easy to follow)

- Figure 1: by "seeding times", do you mean times of importation events? (I would advise to use the same terminology through the text). Also, if ranges reflect the distribution of importation dates, rather than uncertainty around an estimate of seeding times", I'm surprised it doesn't go to April (or even after) for most of the sites (and that's also somehow related to my above comment on CEST estimates)

- Figure 2 is very interesting, but just one question: would it be possible to estimate and report credible intervals for CEST? Minor additional comment on that figure: the authors could consider using colorblind friendly colors as proposed by ColorBrewer (<https://colorbrewer2.org/#type=sequential&scheme=BuGn&n=3>)

Reviewer #2 (Remarks to the Author):

Ragonnet-Cronin et al. explore the effects of NPIs on SARS-CoV-2 transmission and deaths in geographic locations around the world using phylodynamic models. Their results make an important contribution in that they confirm using genomic sequence data earlier reports that delays in implementing NPIs after the virus was introduced into a community resulted in increasingly severe epidemics with more deaths.

There are just a few minor points I hope the authors can address:

How viral introduction times are imputed from phylogenies is never fully explained. Is this the time of the MRCA for a region or the height of particular node identified as an importation event?

"to estimate viral effective population size through time growth rates of effective population size.". This not clear, perhaps missing an 'and'?

Discussion: "at a significance level of 0.001" -- is this a p-value?

On a final note, attributing MCMC convergence (or lack thereof) to violation of model assumptions seems odd to me. Certainly there are ways in which mis-specifying a model could affect mixing/convergence (poor choice of priors, parameterizations leading to poor identifiability or correlations among variables), but it's not clear to me that violations of any particular model assumption mentioned would result in poor MCMC performance.

Reviewer #3 (Remarks to the Author):

The authors investigate evidence from population data on COVID genetic sequence data concerning the effects of public policies (non-pharmaceutical interventions, NPIs) on the pandemic. Understanding testing data has been difficult due to large fluctuations in who gets tested, why, when and how. Genetic sequence data carry some signal about transmission pathways and the size of the latent infection pool, and this signal may be at least somewhat robust to how the sequences were sampled. The authors are experts in this kind of analysis, and there is interest in seeing what they can and cannot properly infer.

A challenge arising is that the genetic data lead relatively directly to estimates of quantities that are not quite the quantities of immediate epidemiological interest. We obtain estimates of "effective sample size" and "central epidemic seeding time (CEST)" when we would rather know about number of infections and age of the epidemic. The authors are well aware of these issues, but nevertheless more could be done to make sure the distinctions remain clear throughout the article.

Some specific comments follow.

1. The authors focus on the first wave of the epidemic, up to May 30, 2020. It may be too much to expect this article to be updated, but perhaps the authors could explain how well the results do, and do not, fit in with experiences since that time.

2. Fig 1 was difficult to interpret on a first read. Partly, this is an unavoidable consequence of the novel measure of CEST and the fact that the results come before the methods. I think the disclaimer in the Fig 2 caption should be moved ahead to Fig 1. Perhaps black dotted vertical lines between regions would help the reader align the bars with the corresponding labels? When I came back to this figure later it seemed clear, so I don't have major concerns about this. Perhaps the important thing to know when reading Fig 1 are what "seeding time" is, and that each observed sequence has its own individual estimated seeding time.

3. CEST measures relative pressure of local growth vs introductions. High introductions relative to local growth give high CEST, i.e. late mean time of ancestral import, ignoring issues of susceptible depletion. Thus, (on page 8), "Sites with larger census population sizes had older central epidemic seeding times" may be an artifact: larger populations should expect to have more within-population transmission relative to between-population transmission.

4. page 8. "CEST was estimated independently of the phylodynamic estimates of viral effective population size" I'm not sure what this means - both were estimated using the same sequences, and the fact that different programs were run is not necessarily protective: imagine using one statistical software program to estimate the mean of a collection of numbers, and another to estimate the median.

5. page 11. start of discussion reads "Among 57 geographical sites sampled across five continents, we found that time from SARS-CoV-2 introduction to time of lockdown (or maximum NPI in locations that never underwent a full lockdown) was associated at a significance level of 0.001 with the severity of the epidemic in each location". Unless I misunderstand, time since introduction is something CEST isn't - but is this comment summarizing the associations with CEST?

6. page 11 line -5. "at the maximum" -> "and the maximum"

7. page 11. The first paragraph of the discussion does not flow clearly in places. Among other things, it is not clear how the lack of correlation of time between tenth case and maximum NPI necessarily implies cryptic transmission. Other explanations are possible. The sentence "This analysis implies..." does not follow on from the previous sentences.

8. page 12. "An association between population size and the number of deaths suggests that the depletion of susceptibles limited onward spread in places with smaller populations." This does not necessarily follow. Smaller places get statistically fewer introductions. In the absence of depletion of susceptibles, one might expect larger places to have more cases, and more ongoing reintroductions once NPIs are in place.

9. Estimating effective sample size, or its relationship to something of direct interest, seems challenging when the population is not closed and we have to take account of multiple introductions. The authors comment (page 13) that the Bayesian phylodynamic skygrowth used to

estimate ESS struggled to converge in some situations. It was not clear to me how these methods dealt with multiple introductions.

10. The subsequent compartment model does explicitly deal with transmission to and from an external global pool. This is probably appropriate for large regions where transmission is primarily self-contained, but remains delicate (to a lesser extent) for small regions where a higher fraction of transmissions may cross in or out across the regional boundary.

11. In Fig 2, the skygrowth ESS seems to be preferred to the compartment model estimate of infection size. Is there a reason for this? Perhaps the authors were not happy with the compartment model analysis for some reason? Superficially, it seems preferable.

12. page 17 line 1. "the time from epidemic origin (CEST)" again suggests we are being asked to view CEST as a time since arrival of the epidemic.

13. A region that locks down very early will have low time between CEST and NPI, and may also expect low cases a month after NPI, but may get hit disproportionately hard by a second wave if the lockdown lifts or compliance wanes. I'm not sure to what extent that is true, but it is at least a valid possible alternative explanation. Perhaps Israel could be an example of this? This suggests to me a limitation of using cases a month after peak NPI as a measure of successful control.

14. Fig S6, second row: If I understand this, it is suggesting that the BEAST model proposes the daily infections have continued to grow in most locations even as the reported cases stabilized. Is this a limitation of the model? Does it have an explanation, such as reduced reporting when the mean age of infection came down?

15. Fig S7: mobility and R_0 both trend down, and I'm not sure a correlation coefficient is a good summary of common trend. I'm sure the p-value for a correlation coefficient is most appropriate when the data look like a scatterplot, which is not the case here. The only time the common trend is not shared is Ahmedabad, which has a negative correlation that is "statistically significant" according to the analysis presented.

REVIEWER COMMENTS

Reviewer #1 (Remarks to the Author):

Preliminary statement: I first need to acknowledge that I consider that I do not have the specific expertise to judge the quality of the multivariate analyses performed in the present study. [For instance, I don't know if an alternative machine learning approach (such as random forest?) could have been more performant and/or relevant than (linear) regression models to analyse the association between epidemic severity and delay between CEST and maximum NPI]

Ragonnet-Cronin et al. here present an interesting study where they used a phylodynamic pipeline to analyse the impact of the timing of non-pharmaceutical interventions in different regions of the world. The study focuses on crucial and timely epidemiological questions. I however have the following concerns regarding the methodology (part of them are actual questions or aspects of the study that were not entirely clear for me):

- I have to admit that I found difficult to follow the pipeline of the phylogenetic and phylodynamic analyses. The authors could consider adding a preliminary Methods paragraph summarising and numerating the main steps that were performed (or, alternatively, generating a figure summarising that workflow). Just some suggestions...

This is an excellent suggestion, and we now include a flow chart summarising the workflow as our first Figure, so that readers can get an overview of the Methods before they read the Results. This figure details the analysis pipeline as well as the reason for inclusion and exclusion for each geographical site in the analysis.

- non-parametric phylodynamic inference: it seems that the skygrowth approach was applied on time-scaled ML trees that were each time inferred from a set of sequences including sequences sampled from the considered site as well as from closed exogenous sequences. Consequently, and except if I missed something in the procedure, such ML tree does not correspond to a single importation event in the considered site. As the authors did, they can be used to identify importation events into the considered site, but a priori not to estimate the trends in the viral effective population size within the site. In other words, I question the application of the skygrowth approach to estimate site-specific viral effective population size based on a ML trees, ML trees that includes several distinct clades gathering sequences sampled in the considered

site, each clade corresponding to a distinct importation event. But again, my apologies if I missed a technical aspect in that step of the pipeline detailed in Supplementary Materials

There are two points to note in response to the reviewer's comment.

1. Exogenous sequences were included in our initial maximum likelihood phylogenies in order to estimate the distribution of times of viral introductions. They were removed from phylogenies before we used those phylogenies to estimate viral effective population size.
2. Nonetheless, the reviewer is correct that as a regional epidemic may result from multiple importation events, the resulting regional clades would not be monophyletic in a larger phylogeny. As a consequence, the estimation of viral effective population size for that region based on a paraphyletic clade as an approximation. This approximation is more problematic early in the epidemic, and becomes less problematic as you approach the present when the majority of transmission events observed are then taking place within the region of focus.

To address the first point above, we have now noted this point in the Methods, l348

“Time-resolved phylogenies were constructed, as detailed above, with all exogenous sequences removed prior to analysis.”.

We now note the limitation of the skygrowth model with regards to the second point above in the Discussion. This limitation is one of the key reasons we also felt it necessary to fit a compartmental model to our data, which explicitly parametrises importations to better estimate the size of local epidemics, l290

“One limitation of the skygrowth phylodynamic model is that it does not explicitly consider imports into the region of interest. As sequences from the region are generally derived from multiple distinct importation events the method may incorrectly estimate viral effective population size for a region, especially early on. However, as an epidemic expands in a region and more infections are attributable to endogenous transmission, the approximations in this method improve. Lineage imports were explicitly parameterized in the SEIJR model and results were consistent across both methodologies. “

- if I followed correctly, the CEST will also be impacted by importation events that could have occurred after the start of NPIs, or even close to it (?). If this is case, I am actually wondering if this is actually the good estimate to be compared with the maximum NPI. It could indeed impact the delay between CEST and maximum NPI that the authors aim at comparing with the epidemic severity. In other words, why not only try to focus on early importation events (e.g., important events before, or at least some time before, the maximum NPI)?

We calculated the distribution of introduction times from the phylogeny for each region, and evaluated the effect of the time delay between viral introduction and the maximum NPI on epidemic severity. We investigated multiple definitions of viral introduction alongside CEST (5th

percentile and 25th percentile of the distribution, previously explained in Supplementary Material, section 1C). We selected the CEST for 2 reasons 1) it normalises for different patterns of viral introduction over time across locations: some places had some very early introductions that did not seed many infections, but others did not 2) it ensures that descendant viruses have contributed to the local epidemic. Of note, in only one case (out of 57) does the CEST fall before the NPI (Abruzzo) and in one case, it fell very close to the maximum NPI (Tel Aviv; dates are shown for all locations in Supplementary Figure 2). Therefore, while we agree that other definitions of delays to lockdown might also fit the data, we think the CEST is at least as good as other metrics. To respond to the reviewer's point, we now make reference to the sensitivity analysis in the Results, l427:

“As a phylogenetic estimate of the time of viral introduction, we use the Central Epidemic Seeding Time (CEST), the mean time of viral introduction weighted by the number of samples descended from each viral introduction. Sensitivity analyses utilising other definitions of viral introduction are presented in the Supplementary Materials. “

We also expand the Results on alternative measures of seeding time in the Supplementary Materials into its own section within the Supplementary Materials (section 4E: Impact of seeding time definitions)

Additional comments and questions:

- model-based phylodynamic inference of epidemic size and reproduction numbers: the SEIJR is very interesting. On question though: why only 50 exogenous sequences? This number seems rather low to estimate the number of independent importation events. I know that it was not the primary purpose of that step of the pipeline but it can probably tend to underestimate the number of distinct importation events that can in turn impact the inference of the epidemic size

We used fifty exogenous sequences encompassing the full time-range of GISAID samples, selected at random, as well as all sequences from GISAID that were ≤ 2 substitutions away from the sequences in the regional dataset (bearing in mind that identical sequences has been removed from the dataset). Therefore the total number of exogenous sequences exceeded 50 in every BEAST run. This is now noted in the Methods, l375.

“Fifty exogenous sequences encompassing the full time-range of GISAID samples were selected each time at random as background, and to these we added sequences from GISAID that were ≤ 2 substitutions away from the sequences in the regional dataset calculated.”

Nonetheless, the reviewer may be correct in their assessment that this is still insufficient to correctly estimate importation rate and we now address this point within the relevant section of the Methods where the phylodynamic model is described, l379:

"It is possible that this number of exogenous sequences is insufficient to correctly estimate import rates into our regions of interest; however this objective was not a focus of the present analysis."

- in my opinion, a sampling map of selected sites (+ the amount of genomic sequences included by site) is missing and could be very informative

We now include a map showing all the locations and the number of sequences from each site as new Sup Figure 2.

- line numbers (as well as figure number and legend below each Supplementary Figure) would have been useful for the reviewing process

We apologise and completely agree. Line numbers are now included,

- my general feeling is that some parts of the Supplementary Materials should move to the main Methods text to improve clarity

We have now moved all methods concerning the estimation of introduction times and the skygrowth model into the main Methods. We include a lot more detail on the BEAST SEIJR model in the main Methods, although the details are still in their own section in the Supplementary Methods. We have kept the details of genetic data cleaning and inclusion/exclusion of study sites in the Supplementary Materials, although a summary of the latter is now included in the new Figure 1.

- Abstract, "The time elapsed between epidemic origin and maximum intervention is strongly associated with different measures of epidemic severity and explains 11% of the variance in reported month after the most stringent intervention": it is only a matter of vocabulary, but I don't know if "strongly associated" is a good match with 11% (?)str

We have removed the word "strongly" from the abstract, l39.

- Introduction, "Comparisons beyond Europe are more complex due to greater variability in epidemiological surveillance and epidemic growth rates": the authors should maybe consider a less Europe-centered version of that sentence ("Comparisons between location from different continents are..." or something like that)

The reason for this choice was not European centrism, but rather because the previous paragraph explains that comparisons within Europe have been possible because reporting

mechanisms are more consistent. Nonetheless, we would not want other readers to draw the same conclusion as the reviewer, and we have changed the phrasing, l87

“International Comparisons are more complex due to greater variability in epidemiological surveillance and epidemic growth rates”

- Discussion, "Notably, the time between detection of the tenth case at each site at the maximum NPI was not predictive of the number of deaths": "..._and_ the maximum NPI...?"

We thank the reviewer for noticing this typo and we have made the change, l238

- Discussion - as acknowledged by the authors, "international comparisons of NPI effectiveness have been complicated by widely varying testing strategies in different locales and most epidemiological models are highly reliant on reported COVID-19 diagnoses and deaths. Our ability to run analyses on such a wide range of locations derives from the fact that our model is parameterised entirely by genetic data": I agree with this statement but, if I'm correct, the authors do however use death counts for measuring epidemic severity. In that context, the authors should probably discuss the impact on their analyses of some countries underreporting COVID-19 mortality, something that was for instance assessed by comparing this mortality to over-mortality estimated by comparing 2020 to previous years (as some journalists of the NY Times did)

This is a really good point, which we insufficiently acknowledge within the Discussion, l263

“Nonetheless, we acknowledge the limitations of reported death data: reliability will vary based on location”

The metric suggested by the reviewer may indeed be less biased but is not available at the level of resolution at which we analysed data. We now note this in the Discussion, l263

“Excess mortality figures may provide less biased estimates, but have not been calculated at the level of resolution required for our study.”

- Methods, "The timing of viral introductions was estimated through time-resolved phylogenetic analysis and parsimony reconstruction": why using a simplistic parsimony reconstruction method instead of one of the discrete diffusion or structured coalescent models implemented in BEAST2?

Our aim with this analysis was to generate a distribution of introduction times into each region of interest. Bayesian phylogeographic models have the advantage of estimating the statistical significance of viral migration events but sample size must be limited for computational reasons and results can be influenced by the choice of sample. This approach has the advantage of working extremely rapidly, while still integrating over phylogenetic uncertainty. We now include this explanation as a note in the methods, l340.

“Compared to more sophisticated phylogeographic models implemented in BEAST³⁸, our method has the advantage of greater computational scalability, while still integrating over phylogenetic uncertainty.”

- Methods, "For sites analysed using the BEAST phylodynamic model, we examined the relationship between mobility data provided by Google (google.com/covid19/mobility, analysis limited to transit stations only) and R_t by calculating Pearson's correlation coefficient": this section needs further detail to be imported from Supplementary Materials

Based on the comments of another Reviewer, we have removed the correlation analysis, keeping only the plots of our two variables. We have added in a lot more detail regarding these plots, I432

“For sites analysed using the BEAST phylodynamic model, we examined the relationship between mobility data provided by Google (google.com/covid19/mobility) and R_t ^{29,31}. Google mobility data measures daily mobility at the sub-region level, in relative deviations from maximum mobility prior to the WHO pandemic declaration. Mobility data were available from the 13th of January 2020 and up until the date of cut-off for genetic data (June 7th). For each site analysed, we plotted both variables for the time period between 13th of January and the date of the last sample available for that site. We limited our analysis to mobility associated with transit stations only (one of six streams). Daily R_t estimates through time were extracted for the same time period.”

- Methods, "We chose to focus our analysis on sites with at least 100 sequences available": "sites" is actually a little bit too vague and should be defined once for all somewhere in the text

The reviewer makes a good point. Our definition of site has been expanded upon at the beginning of methods (I311). We also include a new map that denotes each site and the number of sequences used (new Sup Figure 2).

“The GISAID database classifies the geographical origin of each sequence with four levels of resolution: world region, country, division and location, We chose to focus our analysis on sites, regardless of scale, with at least 100 sequences available (although some sites with fewer sequences were analysed, see Fig 1 and Supplementary Materials for details). Thus a site, as included in our analysis, could represent anything from a neighbourhood (e.g. Manhattan) to a country (e.g. Denmark), depending on the level of resolution of geographical data available with sequences from that region. All sites analysed are displayed in Sup Fig 2 and listed in Sup Table 1. When data were available for sites located within each other (e.g. New Orleans in Louisiana), the smaller geographic unit was preferentially selected to better match the premise of the phylodynamic model (see Supplementary Materials and Discussion).”

- Methods, "This model allowed us to assess the reporting rate for each site": what do you exactly mean by "reporting rate"?

We have extended this sentence to explain, I361

"This model allowed us to assess the reporting rate for each site, the proportion of estimated infections that were diagnosed on each day. "

- Supplementary Materials, "We dropped 80% of sequences collected from the UK after March 15th to reduce bias due to sampling": how did you select the UK sequences to keep in your analyses? This subsampling procedure should be described in detail and motivated

The thought process that led to this decision, and the procedure, have been explained in more detail, Sup Materials, section 1A, I10.

"The Coronavirus Disease Genomics UK Consortium (COG-UK) was launched in March 2020 with the aim of sequencing 10% of all COVID diagnoses. As a result, the proportion of sequences originating from the UK increased dramatically from this date. Disproportional representation of sequences from different places is known to bias phylodynamic and phylogeographic models, thus we dropped 80% of sequences collected from the UK after March 15th. UK sequences were dropped at random."

- Supplementary Materials, "We split our data into subsets (~2,000 sequences)...": did you randomly split your data into subsets?

Yes, we have added this detail to the sentence, Sup Materials, section 1A, I19.

"We split our data randomly into small subsets (~2,000 sequences) to accelerate the cleaning process"

- Supplementary Materials, "... and to each subset added a set of 500 sequences spanning the time period from the first SARS-CoV-2 sample (24/12/2019) to the last sample date": I'm not sure to understand that step (that is the motivation behind it but also how the 500 sequences were selected and the distributed between the 12/24/19 and the last sample date. Could the authors develop further that part?

This procedure formed part of our data cleaning, to eliminate sequences that were either misaligned or that did conform to the molecular clock. Such sequences would hinder time-resolved phylogenetic reconstruction. We have clarified the procedure , Sup Materials, section 1A, I19-24.

“We split our data randomly into small subsets (~2,000 sequences) to accelerate the cleaning process. To each subset we added a set of 500 sequences spanning the time period from the first SARS-CoV-2 sample (24/12/2019) to the last sample date. These latter 500 sequences were used to improve the resolution of time within the subset. Up to 4 sequences were selected at random for each day in the time period from the first SARS-CoV-2 sample (24/12/2019) to the last sample date, totalling 565 unique sequences spanning the time period.”

- Supplementary Materials, "Viral effective population size and the number of infections were estimated from a distribution of trees (generated through maximum likelihood bootstrap and Bayesian inference, respectively)": do you mean that you repeated the analyses while using trees inferred as described in "Estimating the time of regional viral introductions" and then also as described in "Non-parametric phylodynamic inference"? (The connection between the different steps of the pipeline is not always easy to follow)

We now include a flow chart to clarify the pipeline (new Figure 1) and we apologise that our methodology was not clearer. We have also expanded this explanation; Sup Methods, section 1C, I70:

“Viral effective population size through time was extracted from the nonparametric skygrowth model. The number of infections were estimated in the Bayesian SEIJR model in BEAST. In both cases, parameters are estimated from a distribution of trees therefore variance in effective population size and in estimated infections could be calculated directly from those distributions.”

- Figure 1: by "seeding times", do you mean times of importation events? (I would advise to use the same terminology through the text). Also, if ranges reflect the distribution of importation dates, rather than uncertainty around an estimate of seeding times", I'm surprised it doesn't go to April (or even after) for most of the sites (and that's also somehow related to my above comment on CEST estimates)

We have changed “seeding time” to “time of importation event” in the legend of Figure 2 (previously figure 1). Because we found the expression epidemic seeding time to be useful in some phrasings, we have retained both expressions in the text, but we have introduced them both together each time. For example, I137

P6, I13 “Distribution of phylogenetically-inferred seeding times of importation events for the 57 sites included in our analysis, also referred to as “seeding time”

The reviewer is correct, the ranges reflect the distribution of importation times, not the uncertainty, and the estimates do indeed go beyond April. The figure had been cut-off by mistake. We have replaced the figure with one that shows the entire 25th to 75th percentile range (new Figure 2). This is now stated more clearly in the legend, I143

“Boxes represent the interquartile range of the distribution and whiskers indicate the 25th and 75th percentiles of the distributions. Ranges reflect the distribution of importation dates, rather than uncertainty around an estimate of seeding times.”

- Figure 2 is very interesting, but just one question: would it be possible to estimate and report credible intervals for CEST? Minor additional comment on that figure: the authors could consider using colorblind friendly colors as proposed by ColorBrewer (<https://colorbrewer2.org/#type=sequential&scheme=BuGn&n=3>)

We thank the reviewer for both these excellent suggestions. We have modified the figures (new Figure 3 and Sup Fig 2) to now display the 95% CI for CEST estimation. We have modified the colours in both figures to use colour-blind friendly colours.

Reviewer #2 (Remarks to the Author):

Ragonnet-Cronin et al. explore the effects of NPIs on SARS-CoV-2 transmission and deaths in geographic locations around the world using phylodynamic models. Their results make an important contribution in that they confirm using genomic sequence data earlier reports that delays in implementing NPIs after the virus was introduced into a community resulted in increasingly severe epidemics with more deaths.

There are just a few minor points I hope the authors can address:

How viral introduction times are imputed from phylogenies is never fully explained. Is this the time of the MRCA for a region or the height of particular node identified as an importation event?

The reviewer makes a good point, this was not explicitly stated, but we have now done so, l337:

“The time of the importation event is counted as the midpoint along the branch between the exogenous ancestral node and the most recent common ancestor node within the region.”

We also would like to point the reviewer towards the sarscov2 library where all functions can be accessed <https://github.com/emvolz-phylogenetics/sarscov2Rutils>

"to estimate viral effective population size through time growth rates of effective population size.". This not clear, perhaps missing an 'and'?

Thank you for spotting this typo, it has been corrected, l348

"For each region, we applied a *skygrowth* model (16,17) (version 0.3.1) to estimate viral effective population size through time and growth rates of effective population size"

Discussion: "at a significance level of 0.001" -- is this a p-value?

Indeed the significance level indicated the p value, but we have modified the sentence, l232:

"we found that time from SARS-CoV-2 introduction to time of lockdown (or maximum NPI in locations that never underwent a full lockdown) was significantly associated with the severity of the epidemic in each location during the first wave".

On a final note, attributing MCMC convergence (or lack there of) to violation of model assumptions seems odd to me. Certainly there are ways in which mis-specifying a model could affect mixing/convergence (poor choice of priors, parameterizations leading to poor identifiability or correlations among variables), but its not clear that to me that violations of any particular model assumption mentioned would result in poor MCMC performance.

The reviewer is entirely correct, and we now expand our possible explanations for lack of convergence in the Discussion, l274:

"Lack of convergence can occur because of problematic datasets, in which samples may for example not be collected at random; because the model is mis-specified; or because one of the model assumptions is violated, for example the population is not randomly mixing."

Reviewer #3 (Remarks to the Author):

The authors investigate evidence from population data on COVID genetic sequence data concerning the effects of public policies (non-pharmaceutical interventions, NPIs) on the pandemic. Understanding testing data has been difficult due to large fluctuations in who gets tested, why, when and how. Genetic sequence data carry some signal about transmission pathways and the size of the latent infection pool, and this signal may be at least somewhat

robust to how the sequences were sampled. The authors are experts in this kind of analysis, and there is interest in seeing what they can and cannot properly infer.

A challenge arising is that the genetic data lead relatively directly to estimates of quantities that are not quite the quantities of immediate epidemiological interest. We obtain estimates of "effective sample size" and "central epidemic seeding time (CEST)" when we would rather know about number of infections and age of the epidemic. The authors are well aware of these issues, but nevertheless more could be done to make sure the distinctions remain clear throughout the article

We have made many changes to the methods and results which we hope will clarify this distinction for the reader. For example, I167

“As a phylogenetic estimate of the time of viral introduction, we use the Central Epidemic Seeding Time (CEST), the mean time of viral introductions weighted by the number of samples descended from each viral introduction.”

In addition, we emphasize the point in the Introduction, I109

“we reconstruct the epidemic trajectories of SARS-CoV-2 outbreaks for locations across the world for which genetic data were available, to evaluate the scale of epidemics”

We explicitly note that viral effective population size is not directly relatable to the number of infections as a limitation in the discussion, I265

“Viral effective population size is not always a quantity directly relatable to the number of infections⁷; however, our analyses support the use of viral effective population size at maximum NPI as a proxy for SARS-CoV-2 epidemic size at that time”

Some specific comments follow.

1. The authors focus on the first wave of the epidemic, up to May 30, 2020. It may be too much to expect this article to be updated, but perhaps the authors could explain how well the results do, and do not, fit in with experiences since that time.

We thank the reviewer for not requesting that results be updated to new data. We now emphasise that our study focuses on the first wave, by referring to the first wave explicitly throughout the abstract, introduction and discussion. Furthermore, we now conclude, I299

“Our study focused on the first wave of the pandemic; however lifting of interventions and waning compliance have led to subsequent waves of infection in many of the sites included.”

2. Fig 1 was difficult to interpret on a first read. Partly, this is an unavoidable consequence of the novel measure of CEST and the fact that the results come before the methods. I think the disclaimer in the Fig 2 caption should be moved ahead to Fig 1. Perhaps black dotted vertical lines between regions would help the reader align the bars with the corresponding labels? When I came back to this figure later it seemed clear, so I don't have major concerns about this. Perhaps the important thing to know when reading Fig 1 are what "seeding time" is, and that each observed sequence has its own individual estimated seeding time.

We have expanded the legend of the figure (now Figure 2) to briefly summarise the methodology for generating the distribution of importation events and CEST, and included the same disclaimer as in Figure 3, I137-143:

“Ancestral states are reconstructed onto the phylogeny as exogenous or within the region of interest. The midpoint between an ancestral node which is exogenous to the region of interest and a node which is within the region of interest is the time of importation. Local epidemics can be seeded by many importation events. Central box-plot lines represent the central epidemic seeding time (CEST). Note that the CEST does not represent the earliest viral introduction but rather the mean time of viral introduction weighted by the number of samples descended from each viral introduction.”

As suggested by Reviewer 2 above, we have also explicitly stated that seeding time refers to the time of importation events, I136

“Distribution of phylogenetically-inferred times of importation events for the 57 sites included in our analysis, also referred to as “seeding time” “

We experimented with adding lines, but we find that dotted lines make the figure busier and more difficult to decipher.

3. CEST measures relative pressure of local growth vs introductions. High introductions relative to local growth give high CEST, i.e. late mean time of ancestral import, ignoring issues of susceptible depletion. Thus, (on page 8), "Sites with larger census population sizes had older central epidemic seeding times" may be an artifact: larger populations should expect to have more within-population transmission relative to between-population transmission.

We entirely agree, and we conducted this analysis in fact to verify this exact point. As we do not expand any further on this point in the discussion, nor use it to base any of our results, we have removed this result to avoid sowing confusion.

4. page 8. "CEST was estimated independently of the phylodynamic estimates of viral effective population size" I'm not sure what this means - both were estimated using the same sequences, and the fact that different programs were run is not necessarily protective: imagine using one statistical software program to estimate the mean of a collection of numbers, and another to estimate the median.

We were trying to convey the point that the relationship we find between CEST and scale of the epidemic does not result from an analysis where two metrics were co-inferred in a circular way. We make this point more explicit, l209

“). Note that the models for estimating CEST and models for estimating viral effective population size were fit to data independently (although run on the same set of data) and therefore association between these results is not due to a circularity in inference”

5. page 11. start of discussion reads "Among 57 geographical sites sampled across five continents, we found that time from SARS-CoV-2 introduction to time of lockdown (or maximum NPI in locations that never underwent a full lockdown) was associated at a significance level of 0.001 with the severity of the epidemic in each location". Unless I misunderstand, time since introduction is something CEST isn't - but is this comment summarizing the associations with CEST?

CEST is the time at which the majority of imports that have descendants entered the region of interest. While it does not reflect the earliest introduction of the virus in the region, it is the time that is most important in terms of viral introduction. However in our sensitivity analysis, we explored other definitions of the time since viral introduction, using the 5th and 25th percentile of the seeding times distribution as the time of viral introduction and results were consistent (Sup Materials, section 1C). Therefore in our Discussion, these concepts are summarised into “the time since viral introduction”. We now clarify this point at the beginning of our Results section, l168:

“As a phylogenetic estimate of the time of viral introduction, we use the Central Epidemic Seeding Time (CEST), the mean time of viral introduction weighted by the number of samples descended from each viral introduction. Sensitivity analyses utilising other definitions of viral introduction are presented in the Supplementary Materials.”

6. page 11 line -5. "at the maximum" -> "and the maximum"

We thank the reviewer for noting this typo and we have made the correction.

7. page 11. The first paragraph of the discussion does not flow clearly in places. Among other things, it is not clear how the lack of correlation of time between tenth case and maximum NPI necessarily implies cryptic transmission. Other explanations are possible. The sentence "This analysis implies..." does not follow on from the previous sentences.

We have modified the sentence to, I240:

"Based on our numerical analysis, and despite highly variable outcomes across sites, implementing a strong NPI such as national lockdown two weeks earlier would have approximately halved cumulative deaths in the period immediately following lockdown, on average"

8. page 12. "An association between population size and the number of deaths suggests that the depletion of susceptibles limited onward spread in places with smaller populations." This does not necessarily follow. Smaller places get statistically fewer introductions. In the absence of depletion of susceptibles, one might expect larger places to have more cases, and more ongoing reintroductions once NPIs are in place.

We have changed our phrasing to emphasise that this is only one possible explanation, I247.

"An association between population size and the number of deaths could result from the depletion of susceptibles, limiting onward spread in places with smaller populations"

9. Estimating effective sample size, or its relationship to something of direct interest, seems challenging when the population is not closed and we have to take account of multiple introductions. The authors comment (page 13) that the Bayesian phylodynamic skygrowth used to estimate ESS struggled to converge in some situations. It was not clear to me how these methods dealt with multiple introductions.

There is some confusion here, as the issues of convergence were only with the SEIJR model, not the skygrowth model. We have edited the sentence the reviewer to to reflect this, I273

"A limitation of our analysis is that the Bayesian MCMC for our SEIJR phylodynamic model did not converge for all the locations."

The reviewer is correct, the skygrowth phylodynamic model does not account for multiple introductions and that could lead to inaccuracies in the estimation of effective population size, which is why we employed both models where possible. We have now included this limitation to the model in the discussion, I290

“One limitation of the skygrowth phylodynamic model is that it does not explicitly consider imports into the region of interest. As sequences from the region are generally derived from multiple distinct importation events the method may incorrectly estimate viral effective population size for a region, especially early on. However, as an epidemic expands in a region and more infections are attributable to endogenous transmission, the approximations in this method improve. Lineage imports were explicitly parameterized in the SEIJR model and results were consistent across both methodologies”

10. The subsequent compartment model does explicitly deal with transmission to and from an external global pool. This is probably appropriate for large regions where transmission is primarily self-contained, but remains delicate (to a lesser extent) for small regions where a higher fraction of transmissions may cross in or out across the regional boundary.

We acknowledge that import rates will be highly impacted by the size of the region under study, and that for some very small regions with extensive mixing to neighbouring regions, an importation rate estimated in this manner may no longer be meaningful. However, in our simulations, we experimented with different import rates, without the accuracy of reconstructions being affected by those import rates (see Sup Figure 5, simulation 2). The importation rate for each site is estimated directly from the data. We have now made a specific note of this in the Sup Material, section 3, I56:

“The accuracy of our reconstructions was not diminished with higher rates of import (see simulation 2).”

11. In Fig 2, the skygrowth ESS seems to be preferred to the compartment model estimate of infection size. Is there a reason for this? Perhaps the authors were not happy with the compartment model analysis for some reason? Superficially, it seems preferable.

We assume that the reviewer meant Figure 3 (now Figure 4). We chose to show results from the skygrowth model first, and in Figure 3, because it is a simpler model with far fewer parameters, and we obtained results for all sites analysed. We thought it would be simpler for reviewers and readers to understand. Results from the compartmental model are shown in Supplementary Figure 3 and showed the same trend, also significant.

12. page 17 line 1. "the time from epidemic origin (CEST)" again suggests we are being asked to view CEST as a time since arrival of the epidemic.

As mentioned above, CEST is the time at which the majority of imports that have descendants entered the region of interest, which we then refer to as “ the time of viral introduction” and the “time of epidemic origin”. In a sensitivity analysis, we obtained the same findings using

additional definitions for time of introduction (Sup Materials, section 1C). We now clarify this point at the beginning of our Results section, I168:

“As a phylogenetic estimate of the time of viral introduction, we use the Central Epidemic Seeding Time (CEST), the mean time of viral introductions weighted by the number of samples descended from each viral introduction. Sensitivity analyses utilising other definitions of viral introduction are presented in the Supplementary Materials.”

13. A region that locks down very early will have low time between CEST and NPI, and may also expect low cases a month after NPI, but may get hit disproportionately hard by a second wave if the lockdown lifts or compliance wanes. I'm not sure to what extent that is true, but it is at least a valid possible alternative explanation. Perhaps Israel could be an example of this? This suggests to me a limitation of using cases a month after peak NPI as a measure of successful control.

We agree with the reviewer that countries that locked down early and avoided a first wave may fail later if interventions are removed or compliance wanes. Our study was aimed towards looking at the impact of early lockdowns on the first wave of the pandemic, rather than successful control in the long term. We don't think this is a limitation of our study or our model, but given that data is getting dated, we have now made it clearer throughout the paper that we are referring to success in controlling the first wave of the pandemic, by specifically stating we are looking at COVID-19 mortality “during the first wave”. Furthermore, we now conclude, I299

“Our study focused on the first wave of the pandemic; however lifting of interventions and waning compliance have led to subsequent waves of infection in many of the sites included.”

14. Fig S6, second row: If I understand this, it is suggesting that the BEAST model proposes the daily infections have continued to grow in most locations even as the reported cases stabilized. Is this a limitation of the model? Does it have an explanation, such as reduced reporting when the mean age of infection came down?

On one hand, the estimated number of infections at the most recent time points shown on these graphs are actually predictions from the model, rather than reconstructions, because no samples were available beyond a certain time point. We now indicate the last sample available for each site on the graph and make the point in the legend that estimates beyond those points are projections, I479.

“The last sample from each location is indicated by a vertical line of each graph, estimates beyond that time point are projections from the SEIJR model rather than phylodynamic reconstructions. “

Nonetheless, the reporting rate decreases before the last sample in 14/30 locations, and this finding is due to limitations in the SEIJR model's ability to reconstruct recent dynamics when dramatic changes occur close in time to the last sample. We now note this limitation in the Discussion, l282

“Changes in transmission rates close in time to the last sample are difficult to detect using genetic data ¹⁶ and in our SEIJR model, the number of estimated daily infections tended not to decrease or stabilise , despite external evidence that reported cases were levelling off.”

15. Fig S7: mobility and R0 both trend down, and I'm not sure a correlation coefficient is a good summary of common trend. I'm sure the p-value for a correlation coefficient is most appropriate when the data look like a scatterplot, which is not the case here. The only time the common trend is not shared is Ahmedabad, which has a negative correlation that is "statistically significant" according to the analysis presented.

We agree with the reviewer's assessment. The correlation analysis between mobility and our estimated R(t) served to validate our R(t) estimates, but we should not have performed a correlation. The correlation analyses have been removed from the Results and Discussion. The Discussion now states, l269:

“R_t decreased synchronously with reductions in human mobility, as previously demonstrated for R_t estimates derived from traditional epidemiological models”

Sup Fig 6 has been edited to remove the results from the correlation.

REVIEWERS' COMMENTS

Reviewer #1 (Remarks to the Author):

Ragonnet-Cronin addressed all my comments and I thank the authors for the clarity of their answers to my comments and questions. The methodology used and the workflow are now also, in my opinion, much easier to follow. A few additional minor remarks:

- line 40, "indicating that many locations experienced long periods of cryptic transmission": "suggesting that many locations experienced long periods of cryptic transmission"?

- Figure 4: maybe it should be made more explicit in the figure legend that viral effective population size was here extracted at maximum NPI

- lines 303-304, "reinforce previous findings that seroprevalence is far below that needed for herd immunity": this statement deserves maybe more development in the discussion

Reviewer #2 (Remarks to the Author):

My original concerns were minor and the authors have addressed each of them. In general, the manuscript was strong to begin with and the revisions have only strengthened it further.

-David Rasmussen

Reviewer #3 (Remarks to the Author):

The authors have made a thorough revision, and addressed my questions.

Even if the association between the timing of peak non-pharmaceutical interventions for the first wave of COVID-19 is secure, there is scope for variation in interpretation. Perhaps late peak NPI is a result, rather than a cause, of epidemic severity. No observational study can completely avoid such interpretation issues, and the article is sufficiently cautious on the topic (in the title, and elsewhere).

Suppose that two countries with equal introduction times and equal initial NPI responses have different epidemic trajectories, for any other reason. The country with a big outbreak increases its level of NPI (and thus has a late peak NPI) and the country with the small outbreak does not. This may seem like a plausible scenario explaining the observed association. The author's causal interpretation in the discussion (strong lockdowns prevent of the association is also plausible, and intuitively more so. I am not trying to dismiss their finding. I wonder if it would be helpful to explicitly say that they are dismissing the reverse causal explanation as less plausible. Or is there some more concrete reason to dismiss it?

REVIEWERS' COMMENTS

Reviewer #1 (Remarks to the Author):

Ragonnet-Cronin addressed all my comments and I thank the authors for the clarity of their answers to my comments and questions. The methodology used and the workflow are now also, in my opinion, much easier to follow. A few additional minor remarks:

- line 40, "indicating that many locations experienced long periods of cryptic transmission": "suggesting that many locations experienced long periods of cryptic transmission"?

As the abstract has been significantly shortened, this sentence is no longer a part of it.

- Figure 4: maybe it should be made more explicit in the figure legend that viral effective population size was here extracted at maximum NPI

Very good point, we have clarified this point in the legend of Figure 4.

- lines 303-304, "reinforce previous findings that seroprevalence is far below that needed for herd immunity": this statement deserves maybe more development in the discussion

The paragraph l245-255 explains how findings from previous models support or not whether herd immunity has been attained, and in what way our model supports the hypothesis that herd immunity has not been reached. Maybe the reviewer thinks that we have not demonstrated that seroprevalence is FAR below that needed for herd immunity, so we have removed the word "far" from the sentence.

Reviewer #2 (Remarks to the Author):

My original concerns were minor and the authors have addressed each of them. In general, the manuscript was strong to begin with and the revisions have only strengthened it further.

-David Rasmussen

Thank you David!

Reviewer #3 (Remarks to the Author):

The authors have made a thorough revision, and addressed my questions.

Even if the association between the timing of peak non-pharmaceutical interventions for the first wave of COVID-19 is secure, there is scope for variation in interpretation. Perhaps late peak NPI is a result, rather than a cause, of epidemic severity. No observational study can completely avoid such interpretation issues, and the article is sufficiently cautious on the topic (in the title, and elsewhere).

Suppose that two countries with equal introduction times and equal initial NPI responses have different epidemic trajectories, for any other reason. The country with a big outbreak increases its level of NPI (and thus has a late peak NPI) and the country with the small outbreak does not. This may seem like a plausible scenario explaining the observed

association. The author's causal interpretation in the discussion (strong lockdowns prevent of the association is also plausible, and intuitively more so. I am not trying to dismiss their finding. I wonder if it would be helpful to explicitly say that they are dismissing the reverse causal explanation as less plausible. Or is there some more concrete reason to dismiss it?

We agree with the reviewer that no observational study can demonstrate an association, and so we have modified the last paragraph of the discussion to state, 1298:

“In conclusion, we have shown that across five continents, longer delays from viral introduction to lockdowns were associated with more infections at lockdown and more deaths one month after lockdown. The association may be causal but an observational study such as ours cannot draw that conclusion.”